DOI: 10.1038/ncomms11741　　OPEN

# Separating hydrogen and oxygen evolution in alkaline water electrolysis using nickel hydroxide

Long Chen[1], Xiaoli Dong[1], Yonggang Wang[1] & Yongyao Xia[1]

Low-cost alkaline water electrolysis has been considered a sustainable approach to producing hydrogen using renewable energy inputs, but preventing hydrogen/oxygen mixing and efficiently using the instable renewable energy are challenging. Here, using nickel hydroxide as a redox mediator, we decouple the hydrogen and oxygen production in alkaline water electrolysis, which overcomes the gas-mixing issue and may increase the use of renewable energy. In this architecture, the hydrogen production occurs at the cathode by water reduction, and the anodic $Ni(OH)_2$ is simultaneously oxidized into $NiOOH$. The subsequent oxygen production involves a cathodic $NiOOH$ reduction ($NiOOH \rightarrow Ni(OH)_2$) and an anodic $OH^-$ oxidization. Alternatively, the $NiOOH$ formed during hydrogen production can be coupled with a zinc anode to form a $NiOOH$-Zn battery, and its discharge product (that is, $Ni(OH)_2$) can be used to produce hydrogen again. This architecture brings a potential solution to facilitate renewables-to-hydrogen conversion.

[1] Department of Chemistry, Shanghai Key Laboratory of Molecular Catalysis and Innovative Materials, Institute of New Energy, iChEM (Collaborative Innovation Center of Chemistry for Energy Materials), Fudan University, Shanghai 200433, China. Correspondence and requests for materials should be addressed to Y.W. (email: ygwang@fudan.edu.cn) or to Y.X. (email: yyxia@fudan.edu.cn).

Hydrogen has long been considered a promising alternative to unsustainable fossil fuels because it is vital for the production of commodity chemicals such as ammonia and has great potential as a clean-burning fuel[1,2]. However, 90% of the world's hydrogen is currently obtained by the reformation of fossil fuels[3,4], which consumes much energy and is accompanied by serious $CO_2$ emissions. To realize a hydrogen-based economy, hydrogen must be efficiently and sustainably produced[1–8]. Accordingly, the state-of-the-art (photo) electrocatalysts for water splitting are attracting extensive attention[9–13]. Advanced water electrolysis has been considered as one of the most efficient and reliable approaches to producing hydrogen from renewable energy, such as solar, wind and hydropower, for grid-scale energy storage[14–16] because the electrolysis of water at room temperature stands out as a scalable technology, for which the only required inputs are water and energy (in the form of electricity)[17]. Therefore, electrochemical water splitting is attracting extensive attention[18–31]. However, the general application of water electrolysis currently faces a great challenge.

Room-temperature water electrolysis is generally performed under acidic or alkaline conditions. The water electrolysis under acidic condition is performed in an electrolyser with a proton exchange membrane (PEM); thus, it is called PEM water electrolysis[32–35]. Although PEM water electrolysis systems offer several advantages, such as high energy efficiency, a great hydrogen production rate and a compact design, their application remains hampered by the high cost of the catalysts and membranes[36,37]. The acidic environment limits the catalysts for the oxygen-evolving reaction (OER) and hydrogen-evolving reaction (HER) to noble metals. The expensive PEM, which is necessary to prevent the $H_2/O_2$ mixing, also significantly increases the cost. Furthermore, the short durability of the membrane makes PEM electrolysers too expensive for general applications[37,38]. Cronin's group recently developed a new method to split the conventional PEM water electrolysis process into two steps using the polyoxometalate $H_3PMo_{12}O_{40}$ as a buffer for redox equivalents[39,40]. The separate generation of $O_2$ and $H_2$ simplifies gas handling, puts less stringent demands on the membrane of PEM water electrolysis, and potentially reduces cost[41]. Cronin et al. undoubtedly provided an interesting and bright idea for future water electrolysis technology. However, their results[39,40] only focus on PEM water electrolysis (acid condition) and cannot be used in alkaline water electrolysis.

Compared with PEM water electrolysis, alkaline water electrolysis exhibits inherent low-cost characteristic because it can use a non-precious catalyst and a porous separator[36,37]. However, alkaline water electrolysers are difficult to shut down/start up, and their output cannot be quickly ramped because the pressures on the anode and cathode sides of the cell must always be equalized to prevent gas crossover through the porous cell separator[36]. Therefore, it is notably difficult to efficiently use the intermittent and fluctuating power from renewable energy for conventional alkaline water electrolysis. As an alternative to the porous separator, alkaline anion exchange membranes are considered suitable candidate materials that can easily prevent the gas mixing in alkaline water electrolysers[36,42]. However, similar to PEM water electrolysis, the cost and short durability of alkaline anion exchange membranes limit the scalable application. In particular, high-pressure gases in the electrolytic cell aggravate the membrane degradation[40]. Thus, to make renewables-to-hydrogen conversion both practically and economically more attractive, new electrolyser systems must be developed to prevent product gases from mixing over a range of current densities and effectively use the low-cost characteristic of alkaline water electrolysis. It should be a promising solution to separate

$H_2$ production and $O_2$ production. However, the method to separate conventional alkaline water electrolysis into two steps has never been reported.

In the report by Cronin et al.[39], a polyoxometalate-based redox mediator was used as an electron-coupled-proton buffer in an acid solution to decouple the $H_2$ and $O_2$ evolution in PEM water electrolysis. Similarly, it should be a precondition to find a stable electron-coupled-proton buffer in an alkaline solution to separate the $H_2$ and $O_2$ evolution in alkaline water electrolysis. Nickel hydroxide has been widely used in rechargeable batteries using alkaline electrolytes[43]. Its charge and discharge depend on the reversible transformation of $Ni(OH)_2/NiOOH$[44–47], which can be explained as an electron-coupled proton release/storage process (that is, $H^+$ de-intercalation from $Ni(OH)_2$ or intercalation into NiOOH, Supplementary Fig. 1). Therefore, it should be an interesting topic to use nickel hydroxide as an electron-coupled-proton buffer to separate the $H_2$ and $O_2$ evolution in alkaline water electrolysis.

Here, using nickel hydroxide ($Ni(OH)_2/NiOOH$) as a redox mediator, we successfully split the conventional alkaline water electrolysis process into separate $H_2$ and $O_2$ production steps, which well overcomes the gas-mixing issue and increases the use of renewable energy. In this architecture, the $H_2$ production involves the cathodic $H_2O$ reduction and anodic $Ni(OH)_2$ oxidation ($Ni(OH)_2 \rightarrow NiOOH$), and the subsequent $O_2$ production depends on the cathodic NiOOH reduction and anodic $OH^-$ oxidation. In addition, the formed NiOOH during the $H_2$ production can be coupled with a zinc anode to form a NiOOH-Zn battery, which provides an interesting rechargeable system that produces $H_2$ during the electrolysis and delivers energy on the discharge of the NiOOH-Zn battery.

## Results

**Mechanism of the two-step alkaline water electrolysis.** As shown in Fig. 1a, the $H_2$ production (Step 1) involves a cathodic reduction of $H_2O$ on the HER electrode ($H_2O \rightarrow H_2$) and a simultaneous anodic oxidation of the $Ni(OH)_2$ electrode ($Ni(OH)_2 \rightarrow NiOOH$). The subsequent $O_2$ production (Step 2) occurs on the OER electrode by an anodic oxidation of $OH^-$ ($OH^- \rightarrow O_2$), whereas the NiOOH cathode is reduced to $Ni(OH)_2$. This approach leads to a device architecture for the alkaline electrolytic cell with several important advantages. First, the separate generation of $O_2$ and $H_2$ prevents the product gases from mixing over a range of current densities and simplifies the gas handling, which greatly increase the operation flexibility of alkaline electrolytic cells and make them suitable to be driven by sustainable energy (such as solar energy). Second, this device architecture can produce highly pure $H_2$ and $O_2$ with no membrane, which further reduces the cost of the alkaline water electrolysis technology. Third, the separate $H_2$ and $O_2$ productions require different driving voltages (or power inputs), which implies that we can flexibly use sustainable energy (such as solar or wind power) for $H_2$ production or $O_2$ production based on the output variation in these unstable power sources. Finally, the NiOOH that forms during the $H_2$ production (that is, Step 1) can be coupled with a zinc anode to form a NiOOH-Zn battery for energy storage, and its discharge depends on the cathodic reduction of the NiOOH electrode ($NiOOH \rightarrow Ni(OH)_2$) and the anodic oxidation of the zinc electrode ($Zn \rightarrow ZnO_2^{2-}$). Herein, it should be noted that the cathodic reduction potential of NiOOH (0.45 V versus Hg/HgO) is significantly higher than the anodic oxidation potential of zinc ($-1.15$ V versus Hg/HgO) (ref. 48). Therefore, the NiOOH cathode and Zn anode can be coupled to form the NiOOH-Zn battery system that has been commercialized[49–51].) Its discharge product ($Ni(OH)_2$) can be used to produce $H_2$ again, which provides an interesting

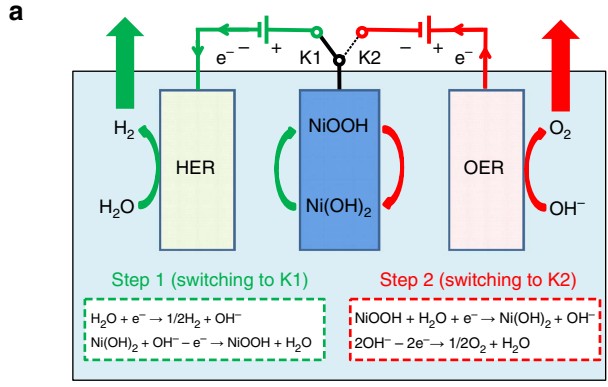

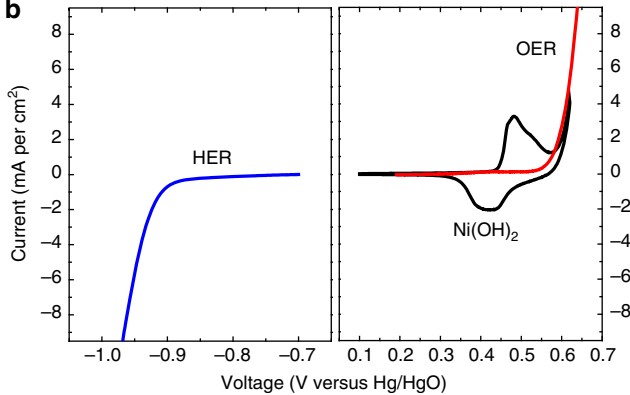

**Figure 1 | Mechanism of the two-step alkaline water electrolysis.**
(**a**) A schematic of the operation mechanism of the cell, where Step 1 (H$_2$ production; switching to K1) involves the cathodic reduction of H$_2$O on the HER electrode (H$_2$O + e$^-$ → 1/2H$_2$ + OH$^-$) and anodic oxidization in the Ni(OH)$_2$ electrode (Ni(OH)$_2$ + OH$^-$ —e$^-$ → NiOOH + H$_2$O). Step 2 (O$_2$ production; switching to K2) includes the cathodic reduction of the NiOOH electrode (NiOOH + H$_2$O + e$^-$ → Ni(OH)$_2$ + OH$^-$) and anodic OH$^-$ oxidization on the OER electrode (2OH$^-$ —2e$^-$ → 1/2O$_2$ + H$_2$O). (**b**) CV curve of the Ni(OH)$_2$ film electrode at a scan rate of 5 mV s$^{-1}$ in 1 M KOH (black line), the linear sweep voltammetric (LSV) data of the commercial RuO$_2$/IrO$_2$-coated Ti-mesh electrode for the OER at a scan rate of 5 mV s$^{-1}$ in 1 M KOH (red line) and the LSV data of the commercial Pt-coated Ti-mesh electrode for the HER in 1 M KOH with a sweep rate of 5 mV s$^{-1}$ (blue line).

rechargeable cycle that produces H$_2$ with the charge (that is, electrolysis in Step 1) and delivers energy with the discharge of the NiOOH-Zn battery.

In this work, nickel hydroxide, which is the conventional electrode material for commercial rechargeable Ni-MH or Ni-Cd batteries, was used as a redox mediator to split the conventional alkaline water electrolysis process into two steps. Before the fabrication of this alkaline water electrolytic cell, the electrochemical profile of Ni(OH)$_2$ in an alkaline electrolyte (1 M KOH) was investigated using a cyclic voltammogram (CV) with a typical three-electrode system, which used a Pt plate and a Hg/HgO electrode as the counter and reference electrodes, respectively. Carbon-nanotube-supported Ni(OH)$_2$ particles were used as the active material to prepare the Ni(OH)$_2$-based film electrode (see the Methods and Supplementary Fig. 2 for details) for the CV measurement, where the carbon nanotube support with high electronic conductivity was only used to alleviate the polarization that arose from the electrode impedance. The CV curve of Ni(OH)$_2$ at a scan rate of 5 mV s$^{-1}$ is shown in Fig. 1b (black line). The OER and HER potentials of the commercial RuO$_2$/IrO$_2$-coated Ti-mesh electrode and Pt-coated Ti-mesh

electrode were also investigated using the three-electrode method for comparison (see the red and blue lines in Fig. 1b). As shown in Fig. 1b, a couple of redox peaks are clearly observed at 0.43 and 0.49 V (versus Hg/HgO) in the CV curve of the Ni(OH)$_2$ electrode because of the reversible cycling between Ni(OH)$_2$ and NiOOH. Obviously, the special potential window for the Ni(OH)$_2$/NiOOH redox couple is located between the onset potential for the OER and the onset potential for the HER. The result indicates that Ni(OH)$_2$ can be used as a redox mediator to split the conventional alkaline water electrolysis process into two steps according to Fig. 1a. The galvanostatic charge-discharge curve of the Ni(OH)$_2$ electrode at a current density of 0.2 A g$^{-1}$ is shown in Supplementary Fig. 3 to clarify the specific capacity of Ni(OH)$_2$ (see the corresponding discussion about Supplementary Fig. 3).

**Performance of the two-step alkaline water electrolysis.** To test the hypothesis in Fig. 1a, an alkaline water electrolytic cell was constructed with a commercial Pt-coated Ti-mesh electrode (Supplementary Fig. 4) for the HER, a commercial RuO$_2$/IrO$_2$-coated Ti-mesh electrode for the OER (Supplementary Fig. 5) and a commercial Ni(OH)$_2$ electrode of conventional Ni-MH or Ni-Cd batteries (Supplementary Fig. 6). The photo profile of the cell is shown in Supplementary Fig. 7, which shows that the Ni(OH)$_2$ electrode (2.5 × 4 cm$^2$) is located between the HER electrode (2.5 × 4 cm$^2$) and the OER electrode (2.5 × 4 cm$^2$). The water electrolysis of the cell was investigated by chronopotentiometry measurements with different applied currents of 100–500 mA. The chronopotentiometry curve (cell voltage versus time) of the electrolytic cell at a constant applied current of 200 mA is shown in Fig. 2a. The chronopotentiometry data of the anode (anodic potential versus time) and cathode (cathodic potential versus time) were also investigated during the electrolysis process and are provided in Fig. 2a. The electrolysis process includes two steps (Steps 1 and 2) with different cell voltages. As shown in Fig. 2a, Step 1 (that is, the H$_2$-production process) exhibits a cell voltage of ~1.6 V, which arises from the difference between the anodic potential of 0.5 V (versus Hg/HgO) of the Ni(OH)$_2$ oxidation (Ni(OH)$_2$ → NiOOH) and the cathodic potential of −1.1 V (versus Hg/HgO) of the H$_2$O reduction (H$_2$O → H$_2$). In Step 2 (that is, the O$_2$-production process), the cell voltage is 0.4 V, which is equal to the potential difference (0.7−0.3 V versus Hg/HgO) between the anodic oxidation of OH$^-$ (OH$^-$ → O$_2$) and the cathodic reduction of NiOOH (NiOOH → Ni(OH)$_2$). In Step 2, the cell voltage sharply increases sharply at the end of electrolysis (Fig. 2a), which indicates that all of the NiOOH has been reduced to Ni(OH)$_2$. In other words, the electrolysis in Step 2 automatically finished after 600 s (= 1,200−600 s), which is equal to the electrolysis time in Step 1 at the identical current of 200 mA. The equal electrolysis time clearly indicates a Coulombic efficiency of ~100%. Photo profiles of the H$_2$ generation in Step 1 and O$_2$ generation in Step 2 are shown in Fig. 2b,c to further characterize the separated steps. In addition, the video evidence also clearly demonstrates the separate H$_2$/O$_2$ generation directly (Supplementary Movies 1 and 2). To clarify the operation flexibility of this electrolyser, the water electrolysis was also investigated at a lower current of 100 mA and a higher current of 500 mA (Supplementary Fig. 8). It should be noted that as a notably mature electrode material, Ni(OH)$_2$ exhibits high efficiency and a long cycle life. These characteristics are also notably important to facilitate the cycle of H$_2$ generation (Step 1) and O$_2$ generation (Step 2). To demonstrate this point, the H$_2$/O$_2$ generation cycle performance was investigated with an applied current of 200 mA. As shown in Fig. 2d, this alkaline electrolytic cell exhibits stable H$_2$ and O$_2$ generation over 20 consecutive cycles. Furthermore,

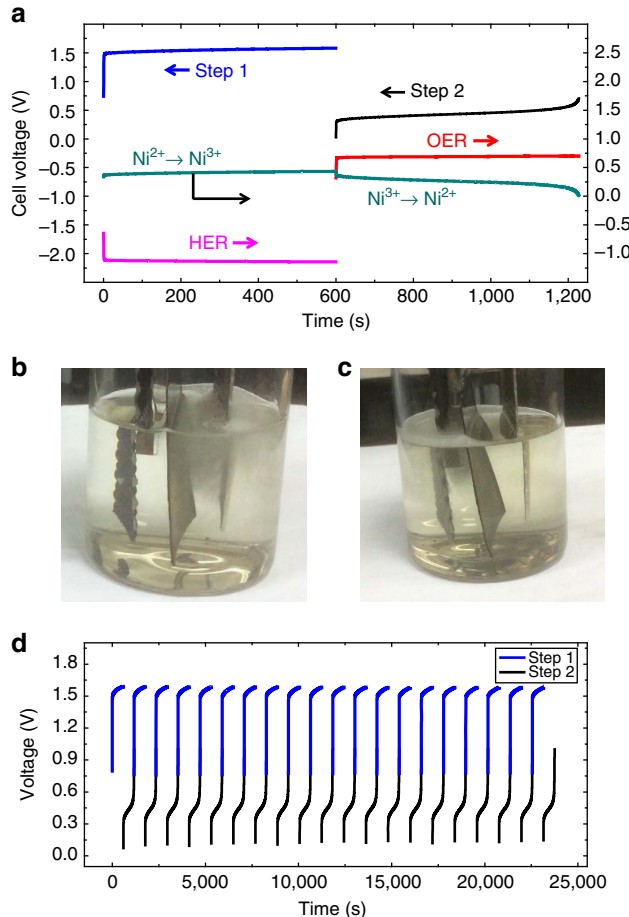

**Figure 2 | Performance of the two-step alkaline water electrolysis.**
(**a**) Chronopotentiometry curve (cell voltage versus time) of the cell at a constant applied current of 200 mA, where the voltages for H$_2$ production (Step 1) and O$_2$ production (Step 2) are labelled by the blue and black lines, respectively. Chronopotentiometry data (potential versus time) of the HER electrode (pink line), Ni(OH)$_2$ electrode (green line) and OER electrode (red line) are provided. ((Voltage of Step 1) = (Potential of Ni$^{2+}$ → Ni$^{3+}$) — (Potential of HER); (Voltage of Step 2) = (Potential of OER) — (Potential of Ni$^{3+}$ → Ni$^{2+}$)). (**b,c**) Photo profiles of the H$_2$/O$_2$ generation in Steps 1 and 2, where it can be detected that H$_2$ and O$_2$ are produced on the HER (**b**) and OER electrodes (**c**), respectively, at separate times (Supplementary Movies 1 and 2 further confirm this point). (**d**) Chronopotentiometry curve (cell voltage versus time) of the H$_2$/O$_2$ generation cycle with an applied current of 200 mA and a step time of 600 s, where the chronopotentiometry data of Step 1 (H$_2$ generation) and Step 2 (O$_2$ generation) are labelled with the blue and black lines, respectively.

100 consecutive cycles of H$_2$/O$_2$ generation are shown in Supplementary Fig. 9 to further demonstrate the stability. As shown in Fig. 2d (or Fig. 2a), the separate H$_2$ production (Step 1) and O$_2$ production (Step 2) require different driving voltages (or power inputs), which implies that we can flexibly use renewable energy, such as solar or wind power, to produce H$_2$ or O$_2$ based on the output variation in these unstable power sources. For example, we can use solar energy at noon to drive the H$_2$-production step, which requires a high driving voltage (or power input), and solar energy at dusk to power the O$_2$-production step, which requires a low driving voltage. The flexibility can increase the use of sustainable energy.

In the above investigation (Fig. 2), a step time of only 10 min (600 s) was used to characterize the separate H$_2$ and O$_2$ production. Such a short time was used to emphasize that we

can flexibly change the operation of our system even within a notably short time. In fact, the electrolysis time in each step can be easily controlled by the applied current. As shown in Supplementary Fig. 10, the electrolysis time in each step can be increased to 12 h with a low current of 20 mA. In addition, the Ni(OH)$_2$ electrode can be cycled with different charge depths (Supplementary Fig. 11). Thus, we can also control the electrolysis time in each step by adjusting the charge depths of the Ni(OH)$_2$ electrode (Supplementary Fig. 12). However, as mentioned in the introduction section, alkaline water electrolysis can use non-precious electrodes for the H$_2$/O$_2$ production[36,37]. Therefore, non-precious electrodes (a Co$_3$O$_4$-based OER electrode and a metal-Ni-foam-based HER electrode) were used to further demonstrate the separate H$_2$ and O$_2$ production (see Supplementary Fig. 13 and Supplementary Movies 3 and 4 for details). Furthermore, according to the previous report by Cronin *et al.*[39], the efficiency of two-step water electrolysis can be evaluated by comparing its total driving voltage (Step 1 + Step 2) to the driving voltage of the corresponding one-step water electrolysis. Therefore, the efficiency of the two-step alkaline water electrolysis using precious or non-precious HER/OER electrodes was calculated according to the method described by Cronin *et al.* (see Supplementary Fig. 14 for detail). As shown in Supplementary Fig. 14a,b, the efficiency of the two-step water electrolysis using precious electrodes (a RuO$_2$/IrO$_2$-coated Ti-mesh electrode for the OER and a Pt-coated Ti-mesh electrode for the HER) is 92% ( = 1.829/1.985) compared with its corresponding one-step water electrolysis. According to the data shown in Supplementary Fig. 14c,d, the efficiency of the two-step water electrolysis using non-precious electrodes (a Co$_3$O$_4$-based electrode for the OER and a metal Ni-foam electrode for the HER) is also ∼92% ( = 1.973/2.137) compared with its corresponding one-step water electrolysis. The achieved efficiency is slightly higher than that (79%) of the two-step PEM water electrolysis reported by Cronin's group[39].

**Purity of the generated H$_2$/O$_2$.** To confirm the purity of the H$_2$/O$_2$ in the separate steps, *in situ* differential electrochemical mass spectrometry was used to measure the gas evolution of the total water electrolysis process at a constant applied current of 200 mA. In this experiment, a quadrupole mass spectrometer with a leak inlet was connected to the alkaline water electrolytic cell with two tubes as the purge/carrier gas inlet and outlet (see the Methods and Supplementary Fig. 15 for details). A pure Ar gas stream was used as the purge gas before electrolysis and the carrier gas during the electrolysis process. Before the online gas analysis, the system was purged with a pure Ar stream for 1 h. The system was further purged with a pure Ar stream for another 1 h, with an online analysis record (Fig. 3a) showing that both O$_2$ and H$_2$ reached a stable background line. Then, the H$_2$-production step (Step 1) was started, and the H$_2$ evolution is clearly observed in the online analysis record. The ion current intensity of O$_2$ obviously remained at the background level in Step 1, which indicates that no O$_2$ was generated in the H$_2$ production process of 30 min (Fig. 3a,b). After the H$_2$ production (Step 1) finished, a rest step of 130 min was performed with a pure Ar stream to eliminate remnant H$_2$ in the system, and a hysteresis of H$_2$ could be observed in the online analysis record. Afterward, the O$_2$-production step (Step 2) was started. As shown in Fig. 3b, the O$_2$ production automatically finished with a total electrolysis time of ∼30 min (1,700 s), which is close to the H$_2$ production time (1,800 s). The minor difference of 100 s may be because of the slight self-discharge of NiOOH in the rest step. However, this description does not indicate that the self-discharge of the NiOOH electrode will be significantly aggravated with a longer rest time (see Supplementary Fig. 16 for an extended

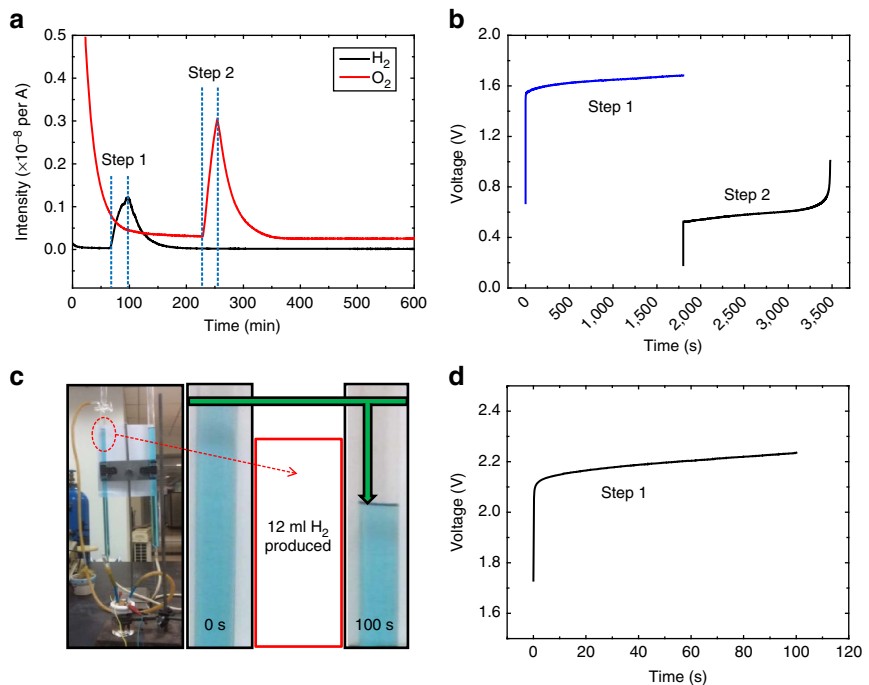

**Figure 3 | Purity of the generated H₂/O₂.** (**a**) *In situ* differential electrochemical mass spectrometry (DEMS) curves of the $H_2$ evolution (black line) and $O_2$ evolution (red line) in the total water electrolysis process at a constant applied current of 200 mA. A pure Ar gas stream was used as a purge gas before electrolysis and the carrier gas in the total electrolysis process. There is a rest time of 130 min between Steps 1 and 2. (**b**) The chronopotentiometry curve (cell voltage versus time) of the electrolytic cell corresponds to the *in situ* DEMS test with two steps: $H_2$-production process of 30 min (Step 1, the blue line) and $O_2$-production process (Step 2, black line). (**c**) A typical apparatus configuration to determine the evolution of the $H_2$ volumes. (The $H_2$ production rate (ml s$^{-1}$) was measured with an applied current of 1,000 mA for 100 s.) Approximately 12 ml $H_2$ was generated after 100 s of electrolysis, as indicated by the green arrow. (**d**) The chronopotentiometry curve (cell voltage versus time) of the electrolytic cell corresponds to the $H_2$-production volume test with an applied current of 1,000 mA for 100 s.

discussion about the self-discharge of the nickel hydroxide electrode). As shown in Fig. 3a, there is no $H_2$ evolution in the $O_2$ production process. Therefore, the results in Fig. 3a,b well demonstrate the purity of the $H_2/O_2$ in the separate steps. Herein, it should be noted that the online gas analysis in our experiment was only used to characterize the purity of the $H_2$ and $O_2$ in separate steps. A typical drainage method (Supplementary Fig. 17) was used to quantify the $H_2$ generation over a specific time length. In this experiment, the $H_2$ production rate (ml s$^{-1}$) was measured with an applied current of 1,000 mA for 100 s (Fig. 3c,d). Figure 3c,d shows that $\sim 12$ ml $H_2$ was generated in the 100 s electrolysis, which is close to the theoretical value (12.67 ml). Therefore, the Faradaic efficiency is 94.7% (12/12.67). In theory, the Faradaic efficiency should be 100%, but the impedance and dissolution of $H_2$ in the aqueous solution may slightly reduce the efficiency. This method was used to measure the generated $O_2$ volume in Step 2 at the identical current of 1,000 mA. The obtained result indicates that $\sim 6$ ml $O_2$ was generated in Step 2. Therefore, the $H_2$-to-$O_2$ ratio is 2:1 in the consecutive cycle of Steps 1 and 2.

**Combination between the H₂-production and NiOOH-Zn battery.** Interestingly, the aforementioned $O_2$-production step (Step 2) can be replaced by the discharge step of the NiOOH-Zn battery (Step 2′), which will enable the coupling of $H_2$ production with a discharge step of the NiOOH-Zn battery (Fig. 4a). As shown in Fig. 4a, the $H_2$ production (Step 1) includes the cathodic reduction of $H_2O$ on the HER electrode ($H_2O \rightarrow H_2$) and the anodic oxidization of the $Ni(OH)_2$ electrode ($Ni(OH)_2 \rightarrow$ NiOOH). Next, the NiOOH electrode that is formed in Step 1 is coupled with a zinc anode to form a NiOOH-Zn battery. The

subsequent discharge step (Step 2′) of the NiOOH-Zn battery is based on the cathodic reduction of the NiOOH electrode (NiOOH $\rightarrow$ Ni(OH)$_2$) and the anodic oxidization (Zn $\rightarrow$ ZnO$_2^{2-}$) of zinc[48–50]. In other words, the architecture in Fig. 4a provides an interesting rechargeable cycle that produces $H_2$ with charge (that is, electrolysis in Step 1) and delivers energy with the discharge of the NiOOH-Zn battery (Step 2′). To confirm this hypothesis, the NiOOH electrode, which formed after the electrolysis for $H_2$ production with an applied current of 200 mA for 600 s, was directly coupled with a zinc-plate electrode in the electrolysis cell to construct a NiOOH-Zn battery. The discharge profile of the NiOOH-Zn battery was investigated with a current of 200 mA. As shown in Fig. 4b, the NiOOH-Zn battery displays a discharge voltage of $\sim 1.6$ V with a total discharge time of $\sim 600$ s. Furthermore, the consecutive $H_2$-production step (Step 1) and discharge step (Step 2′) of the NiOOH-Zn battery can be cycled exactly like a rechargeable battery (inset of Fig. 4b). Further cycle data of Step 1 ($H_2$-production step) and Step 2′ (discharge of the NiOOH-Zn battery) are provided in Supplementary Fig. 18. Therefore, the architecture in Fig. 4a introduces a new energy storage/ conversion approach, in which solar energy can be used to drive the electrolysis (charge process) of Step 1 to produce $H_2$ during the daytime, and the discharge step (Step 2′) of the NiOOH-Zn battery can be used to deliver energy to power electronic devices overnight. To clarify this point for the layperson reader, the NiOOH-Zn battery that formed after the $H_2$-production step was used to power a 1.6 V electric fan (Supplementary Movie 5). The cycle of Steps 1 and 2′ also increases the $Zn(OH)_2$ concentration in the alkaline electrolyte. The $Zn(OH)_2$ alkaline solution can be used to produce $O_2$ and

**a**

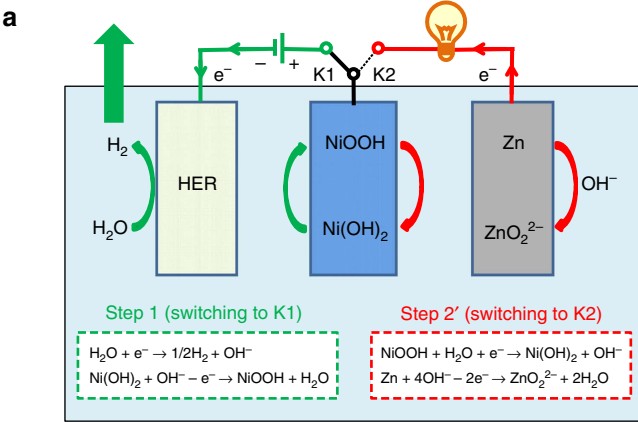

**b**

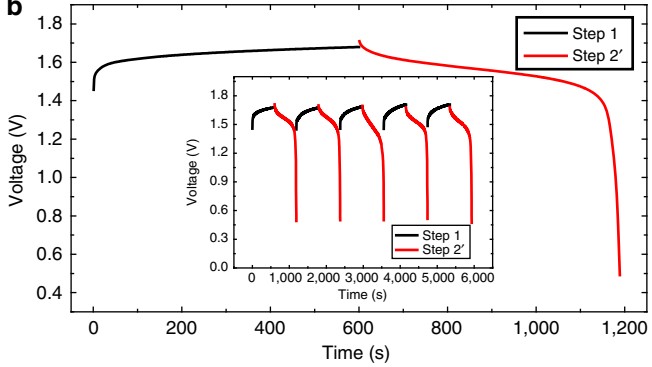

**Figure 4 | Combination of the H2-production and the NiOOH-Zn battery.**
(**a**) A schematic of the operation mechanism of the cell. (Herein, the
H$_2$-production process (Step 1; switching to K1) includes the cathodic
reduction of H$_2$O on the HER electrode (H$_2$O + e$^-$ → 1/2H$_2$ + OH$^-$) and
the anodic oxidization of the Ni(OH)$_2$ electrode (Ni(OH)$_2$ + OH$^-$ — e$^-$ →
NiOOH + H$_2$O). Then, by switching to K2, the NiOOH electrode that
was formed during Step 1 is coupled with a zinc anode to form a
NiOOH-Zn battery, and its discharge (Step 2′) depends on the cathodic
reduction of the NiOOH electrode (NiOOH + H$_2$O + e$^-$ → Ni(OH)$_2$ + OH$^-$)
and the anodic oxidization of the zinc electrode (Zn + 4OH$^-$ — 2e$^-$ →
ZnO$_2^{2-}$ + 2H$_2$O)). (**b**) Chronopotentiometry curve (cell voltage versus
time) of the H$_2$-production process (Step 1, black line) and the discharge
curve of the NiOOH-Zn battery (Step 2′, red line). The electrolysis for H$_2$
production applied a current of 200 mA for 600 s; then, the discharge
profile of the NiOOH-Zn battery was also investigated with a current of
200 mA. The inset is the cycle performance of the H$_2$-production step
(black line) and discharge step (red line) of the NiOOH-Zn battery with an
applied current of 200 mA. As shown in **a**,**b**, Step 1 requires a power input
to produce H$_2$ gas, whereas Step 2′ (discharge of the NiOOH-Zn battery)
can deliver energy to power other devices.

metallic Zn through electrolysis at the proper time with other
energy inputs, such as at night-time with wind power, nuclear
fission and so on. In addition, the rechargeable system based on
the H$_2$ production (Step 1) and discharge of the NiOOH-Zn
battery (Step 2′) exhibits a theoretical energy density of
280 Wh kg$^{-1}$ (see Supplementary Fig. 19 for details), which is
close to the theoretical energy density of conventional Ni-MH
batteries, Ni-Cd batteries or Ni-Zn batteries and higher than the
theoretical energy density of lead-acid batteries and aqueous
Li-ion batteries[51].

## Discussion

We successfully split the conventional alkaline electrolysis into
two steps by using nickel hydroxide as a recyclable redox
mediator. The separate H$_2$ and O$_2$ production overcomes the

challenge of H$_2$/O$_2$ mixing and facilitates the operation of alkaline
electrolysis even with unstable power inputs. The separate H$_2$ and
O$_2$ productions require different driving voltages (or input
power), which implies that we can flexibly use sustainable energy,
such as solar or wind power, with higher efficiency. Finally, the
combination of H$_2$ production and discharge of the NiOOH-Zn
battery potentially provides a new energy storage/conversion
approach for human life. It should also be noted that the
electrochemical redox process of Ni(OH)$_2$/NiOOH is generally
limited by the proton diffusion in the crystalline framework of
Ni(OH)$_2$ or NiOOH[52], which limits the electrolysis rate.
Therefore, solar energy, with the characteristic of low power
loads, should be suitable to drive the new type alkaline electrolytic
cell. However, much effort has been made to develop high-rate
Ni(OH)$_2$ electrodes to increase the power density of Ni(OH)$_2$-
based batteries, which may further enhance the electrolysis rate of
the new type alkaline electrolytic cell. The data in the result
section were obtained using a one-compartment water electrolytic
cell, where all three electrodes (HER cathode, OER anode and
Ni(OH)$_2$ electrode) were immersed in the same electrolyte. In
practical applications, the two-step alkaline water electrolysis can
also be operated with two separate rooms, where the nickel
hydroxide (Ni(OH)$_2$/NiOOH) electrode is used as 'a solid-state
proton buffer' (see the proton buffer mechanism in
Supplementary Fig. 1), which can be moved between room 1
for H$_2$ production (Step 1) and room 2 for O$_2$ production (Step 2;
see the extended discussion in Supplementary Figs 20 and 21 and
Supplementary Movies 6 and 7).

## Methods

**Synthesis and characterization of Ni(OH)$_2$/MWNT composite.** First, multi-
walled carbon nanotubes (MWNTs) underwent a hydrophilic treatment before use.
The MWNTs were sonicated in 30% HNO$_3$ solution for 30 min, filtered and
washed with distilled water, and finally dried at 100 °C for 12 h. According to our
previous report, Ni(OH)$_2$/MWNT composites were prepared by loading Ni(OH)$_2$
on the treated MWNTs in an alkaline medium: 0.3 g of accurately weighed
MWNTs was immersed and dispersed in a bath that contained 2.2 g of 0.1 M
Ni(NO$_3$)$_2$ · 6H$_2$O solution. Then, 0.1 M KOH solution was dropped into the bath
while stirring until the pH of the aqueous solution was 8.5. The resulted product
was filtered and repeatedly washed with distilled water, dried at 80 °C and
weighed. The 70 wt % mass load of Ni(OH)$_2$ in the composites was evaluated by
calculating the weight difference of MWNTs. The Ni(OH)$_2$/MWNT composite was
characterized using S-4800 scanning electron microscopes and a Joel JEM2011
transmission electron microscope.

**Preparation and electrochemical test of the electrode.** The Ni(OH)$_2$/MWNT
composite electrode was prepared according to the following steps: a mixture
of 85 wt% Ni(OH)$_2$/MWNT composites, 10 wt% acetylene black and 5 wt%
polytetrafluoroethylene was thoroughly mixed to form a film, which was pressed
onto a nickel grid (1.2 × 10$^7$ Pa) that served as a current collector surface (1 cm$^2$).
The Ni(OH)$_2$/MWNT composite electrode was characterized by CV with a scan
rate of 5 mV s$^{-1}$ and a galvanostatic charge-discharge test at a current density of
0.2 A g$^{-1}$. The electrolyte was 1 M KOH solution. The onset potential of the OER
on the commercial RuO$_2$/IrO$_2$-coated Ti-mesh electrode and onset potential of
the HER on the commercial Pt-coated Ti-mesh electrode in 1 M KOH were
investigated by linear sweep voltammetric measurements with a sweep rate of
5 mV s$^{-1}$ in 1 M KOH. The aforementioned experiments were performed with
a typical three-electrode method, in which a Pt plate and Hg/HgO (0.098 V
versus the standard hydrogen electrode) were used as the counter and reference
electrodes, respectively. All electrochemical measurements were performed
with a CHI 660D electrochemistry workstation.

**Fabrication of the new type electrolytic cell.** The cell was constructed with a
commercial Pt-coated Ti-mesh electrode (Supplementary Fig. 4) for the HER,
a commercial RuO$_2$/IrO$_2$-coated Ti-mesh electrode for the OER (Supplementary
Fig. 5) and a commercial Ni(OH)$_2$ electrode of conventional Ni-MH or Ni-Cd
batteries (Supplementary Fig. 6). The photo profile of the cell is shown in
Supplementary Fig. 7, where the Ni(OH)$_2$ electrode (2.5 × 4 cm$^2$) is located
between the HER electrode (2.5 × 4 cm$^2$) and OER electrode (2.5 × 4 cm$^2$).

**Water electrolysis investigation.** The water electrolysis of the new type alkaline
water electrolysis cell was investigated using chronopotentiometry measurements

with applied currents of 100, 200 and 500 mA. In Step 1, the HER electrode (that is, Pt coated Ti-mesh electrode) and Ni(OH)$_2$ electrode were connected to the cathode and anode, respectively, of the power source for electrolysis. The duration time of Step 1 (that is, the H$_2$-production step) is 600 s with an applied current of 100, 200 or 500 mA. After Step 1, Step 2 (that is, the O$_2$-production step) was started. In Step 2, the NiOOH electrode, which formed during Step 1, and the OER electrode (that is, the RuO$_2$/IrO$_2$-coated Ti-mesh electrode) were connected to the cathode and anode, respectively, of the power source for the electrolysis. Step 2 automatically stopped when the cell voltage sharply increased at the end of the electrolysis, which indicated that all the NiOOH was converted into Ni(OH)$_2$. The cell voltages ($V$ versus time) of Steps 1 and 2 were recorded to characterize the electrolysis profile. With an additional reference electrode (that is, the Hg/HgO electrode), the chronopotentiometry data (potential versus time) of a single electrode (the HER electrode, Ni(OH)$_2$ electrode or OER electrode) were also recorded during Steps 1 and 2. A schematic illustration of the voltage (or potential) record is provided in Supplementary Fig. 22. As shown in Supplementary Fig. 22, $V_1$ is the cell voltage in Step 1; $V_2$ is the cell voltage in Step 2; $P_1$ is the potential of the HER electrode in Step 1; $P_2$ is the potential of the Ni(OH)$_2$ electrode in Steps 1 and 2 and $P_3$ is the potential of the OER in Step 2. A PARSTAT MC multi-channel workstation was used to perform the water electrolysis investigation of Fig. 2.

**In situ differential electrochemical mass spectrometry.** A quadrupole mass spectrometer (NETZSCH QMS 403 C) with a leak inlet was applied to measure the gas evolution of the total water electrolysis process at a constant applied current of 200 mA. The electrolysis process was performed using a Solarton Instrument Model 1287 electrochemical interface. As shown in Supplementary Fig. 15, the mass spectrometer is connected to the new-type alkaline water electrolysis cell with two tubes as the purge/carrier gas inlet and outlet. A pure Ar gas stream was used as the purge gas before electrolysis and the carrier gas during the electrolysis process. The gas flows were typically 10 ml min$^{-1}$. Before the online gas analysis, the system was purged with a pure Ar stream for 1 h. The system was further purged with a pure Ar stream for another 1 h before Step 1 began. The duration time of Step 1 was 30 min. After the end of Step 1, a rest step of 130 min was performed with a pure Ar stream to eliminate remnant H$_2$ in the system. Then, Step 2 was started and continued until the cell voltage sharply increased. The experiment completed when all remaining O$_2$ in the electrolysis cell was removed.

**Data availability.** All relevant data are available from the authors.

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

## Acknowledgements

We acknowledge funding support from the Natural Science Foundation of China (21333002, 21373060), and Shanghai Science and Technology Committee (13JC1407900).

## Author contributions

Y.W. conceived and designed the experiments. Y.W. and Y.X. directed the project. L.C., X. D. and Y. W. performed the experiments. L. C. and Y. W. co-wrote the paper. All authors discussed the results and commented on the manuscript.

## Additional information

**Competing financial interests:** The authors declare no competing financial interests.

