## [Peer review file · Nature Communications]

Reviewers' Comments:

Reviewer #1 (Remarks to the Author)

This is an interesting result but I don't see the significance. As the authors said, Cronin already suggested this new approach and presented data exploring the technoeconomics. This work suggests that alkaline systems could be more sustainable but then shows cells with precious metal electrodes. Also no efficiencies are given, not power-based, or faradaic i.e. showing how much charge is passed turns into hydrogen. The energy density, or carrying density of the liquid is not discussed.

In summary this work has potential but is far too premature for publication.

Reviewer #2 (Remarks to the Author)

Chen et al. discuss an intriguing concept of combining an electrolyzer electrode with a Ni(OH)₂ battery electrode to generate hydrogen gas without oxygen evolution. Instead, the battery material is charged. Using DEMS, it is demonstrated that hydrogen gas is indeed formed and that the charged battery material can either be used to generate oxygen (using a suitable electrolyzer anode) or electricity (using a Zn cathode). The idea of separating hydrogen and oxygen evolution was pioneered by Cronin, Symes and coworkers (as also cited in the manuscript). Chen et al. applied it successfully to more conventional electrochemical system where commercial electrodes were employed and a simple alkaline electrolyte. The combination of an electrolyzer anode with a Zn cathode is a welcome further (and original) application of the concept. As such, the manuscript should attract considerable attention in energy research. Therefore, the manuscript is suitable for Nature Communications. However, several points need to be addressed before publication can be recommended:

- 1) The charge/discharge times of the battery electrode are 10 min.
 - a) This seems too short to buffer fluctuations from renewable sources, particular the day/night cycle mentioned several times. Please comment on this for the reader.
 - b) Does the device still work efficiently if the battery material is only charged partially (e.g. 1/4, 1/2, 3/4) and discharged partially (e.g. same fractions)? This mode of operation seems highly relevant to the proposed coupling to renewables with arbitrary fluctuations.
- 2) The authors mention self-discharge of NiOOH as the reason for the current difference between hydrogen and oxygen evolution during DEMS.
 - a) This would be serious problem for the envisioned application. Please comment on this, preferably also in the discussion section of the manuscript.
 - b) Self discharge also means that the device could be operated during the day and self discharge overnight? How long would it take to self-discharge?
- 3) P10. "The data of DEMS (i.e. ion current vs. time) can not be directly employed to evaluate the amount of H₂ and O₂."
 - a) Why did your engineer not recommend to perform a calibration for these two gases? See, e.g. <http://mediatum.ub.tum.de/doc/1007122/1007122.pdf>
It would strengthen the manuscript if the authors can show that the ratio between H₂ and O₂ at the end of the experiments is 2:1 as expected and observed in traditional electrolysis.
 - b) The following sentence states that 12 mL H₂ was produced. How was this obtained if not from DEMS? Please include the calculation in the Supplemental Document. What is the expected theoretical value for this calculation?
- 4) For H₂/O₂ operation, 20 cycles are shown in in Fig. 2 and for H₂/Zn(OH)₂ operation, 5 cycles.

Were the operating modes tested by cycling further? These cycle numbers are too low to estimate usefulness and claim the battery is "rechargeable".

5) How does the authors work compare to previous work of the Cronin group? Is it possible to compare efficiencies?

6) The title is not very clear

a) Depending on the mode, the active material is either Ni(OH)₂ or NiOOH, while only Ni(OH)₂ is mentioned in the title. "Nickel hydroxide" might be better suited.

b) H₂ production is also energy storage (in the H-H bond)

c) Splitting alkaline water electrolysis is not very clear either. Separating hydrogen and oxygen evolution is a much better description.

7) References

a) Why are 9 references (ref. 5-13) necessary to support the sentence "hydrogen must be produced in an efficient and sustainable manner"? Isn't this covered sufficiently by the reviews in refs. 1-8? Perhaps, the original research manuscripts (9-13) could be introduced as state-of-the-art (photo)electrocatalysts.

b) The Ni(OH)₂/NiOOH electrode is central to this manuscript, yet no reviews regarding its chemistry are cited (ref. 44 seems to discuss a device). In particular, the "Bode scheme" should be mentioned in the introduction to support the transformation from Ni(OH)₂ to NiOOH. Alternatively, XRD spectra could be provided to support the transformation.

c) There should be a suitable reference on P12 to support that proton diffusion is limiting the redox processes in Ni(OH)₂/NiOOH for non-expert readers.

8) Language

While the manuscript is overall understandable, the language and grammar should be improved prior to publication. Some formulations are unclear, e.g. P4: "By turning into using formed NiOOH as cathode". Is there something missing after into?

Typos (incomplete list):

Several instances (e.g. caption of Fig. 1 and the abstract): produciton -> production

P6. That using a Pt plate -> that used a Pt plate

P6. CNTs supporter -> CNT support

P9. use solar energy at noon to driven H₂-production step -> use solar energy at noon to drive H₂ production

P10. no any H₂ evolution -> no H₂ evolution

P11. Popular reader -> layperson reader

P13. Microscropes -> microscopes

P15. Solution -> Solatron

Reviewer #3 (Remarks to the Author)

The submitted manuscript by Long Chen et al describes an interesting strategy of utilizing Ni(OH)₂ as a redox mediator to decouple the HER and OER of water splitting electrolysis. This manuscript is more like an engineering article rather than a fundamental science work. Therefore, this referees has the following two primary concerns from an engineering perspective.

It seems like in order to directly employ a Ni(OH)₂ electrode as the redox mediator, all the three electrodes (HER cathode, OER anode, and Ni(OH)₂ electrode) are immersed in the same electrolyte, therefore it is a one-compartment water electrolytic cell. Although the authors used online mass spectrometer to confirm the purity of the produced H₂ and O₂, in practical application, it is hard to imagine purging the electrolyte constantly to remove H₂ and O₂ completely.

Therefore, even if H₂ and O₂ are produced separately and consecutively, there still will be residual H₂ and/or O₂ left in the headspace of the water electrolytic cell. Hence, the potential H₂/O₂ mixing issue remains. The authors are recommended to address this point.

Secondly, the stability of the in situ generated NiOOH during HER is crucial to the success of this new design of water electrolysis. As the authors implied in the text, there is a possibility that NiOOH will self discharge. Then the questions are what the capacity of the a typical Ni(OH)₂ electrode and how long it can sustain its charge before discharge, especially in a strongly alkaline aqueous electrolyte?

In summary, a major revision is recommended before further consideration.

Response to Reviewer #1

Comment: This is an interesting result but I don't see the significance. As the authors said, Cronin already suggested this new approach and presented data exploring the technoeconomics. This work suggests that alkaline systems could be more sustainable but then shows cells with precious metal electrodes. Also no efficiencies are given, not power-based, or faradayic i.e. showing how much charge is passed turns into hydrogen. The energy density, or carrying density of the liquid is not discussed. In summary this work has potential but is far too premature for publication.

Response: Thank you very much for kindly reviewing our manuscript and giving a lot of important suggestions. For each question pointed out by you, we would like to answer separately and revise our manuscript according to your suggestions [Please see **Points-1** (significance description), **2** (new data with non-precious electrodes), **3** (efficiency) and **4** (energy density) for detail. In addition, a lot of new data according to the suggestions of reviewer # 2 and 3 (Figures ^{answer} 4-12) was also added in the revised manuscript. See our response to them for detail.]

Point-1: The significance about present work:

As pointed out by you, we have described Cronin *et al.*'s report (*ref. 39,40*) in the introduction section. It is undoubted that Cronin *et al.* provided an interesting and bright idea for future water electrolysis technology. However, their results (*ref.39,40*) only focus on PEM water electrolysis (acid condition), and cannot be used in alkaline water electrolysis. [This sentence highlighted by red underline have been given in introduction to emphasize previous work, and to clarify the relationship between Cronin *et al.*'s reports and present work. Please see revised manuscript (Page 3-4): sentences highlighted by yellow background]. The significance about present work is given as following:

Alkaline water electrolysis has been widely deployed for several decades in large-scale hydrogen production because it can employ non-precious catalyst and porous separator (See *Ref. 36,37*). However, alkaline water electrolyzers are difficult to shut down/start up and their output cannot be ramped quickly because the pressure on the anode and cathode sides of the cell must be equalized at all times to prevent gas crossover through the porous cell separator (*Ref. 36,37*). There is thus a need to develop new approach that can prevent product gases from mixing over a range of current densities and that makes more effective use of the low cost characteristic of alkaline water electrolysis, in order to make renewables-to-hydrogen conversion both practically and economically more attractive. Herein, it should be a promising solution to separate H₂ production and O₂ production. However, how to separate conventional alkaline water electrolysis into two steps, it is never reported up to present. Therefore, the significance of present work can be further summarized as:

- (1) It should be the first time that conventional alkaline water electrolysis was split into separate H₂ production and O₂ production steps by using Ni(OH)₂/NiOOH as a recyclable redox mediator, which potentially increases the operation flexibility of alkaline electrolytic cells. [See Fig.1, Fig.2, Fig.3].
- (2) Using Ni(OH)₂/NiOOH as the recyclable redox mediator, we also combine H₂ production and discharge of NiOOH-Zn battery to provide an interesting rechargeable cycle that can produce H₂ on charge (i.e. electrolysis in step 1) and deliver energy on discharge of NiOOH-Zn battery [See Fig. 4, Movie S5]. The device architecture might provide a new type energy storage/conversion approach for human life. For example, we can

use solar energy to produce H₂ at daytime, and then use NiOOH-Zn battery to power various electronic devices at night time.

- (3) In present work, the alkaline water electrolytic cell can separately produce pure H₂ and O₂ without any membrane [Please See Fig.3, Movie S1, and Movie S2].
- (4) The alkaline water electrolytic cell can also separately produce H₂ and O₂ with non-precious catalyst. [We have employed non-precious electrode to realize separate H₂ and O₂ production (See Point-2 or revised manuscript for detail).] In addition, the Ni(OH)₂/NiOOH also is low cost material that is widely used in rechargeable battery.

Point-2: Separate H₂/O₂ production with non-precious electrodes has been shown in the revised manuscript.

As correctly pointed out by you, it has been mentioned in the introduction section that alkaline water electrolysis systems can employ non-precious catalyst [Corresponding citation Ref. 36: *J. Am. Chem. Soc.* 134(2012)9054-9057; Ref.37: *Progress in Energy and Combustion Science* 36(2010)307-326]. **Obviously, the main topic of present work is to introduce “how to separate H₂ and O₂ production in alkaline water electrolysis by using Ni(OH)₂/NiOOH redox mediator”, rather than catalysts.** Accordingly, we employed the commercialized precious metal electrodes (Pt coated Ti-mesh and RuO₂/IrO₂ coated Ti-mesh) to clarify the separate steps. **However, your suggestion is important. As a result, we have investigated the separate steps with non-precious electrodes in the revised process. The achieved performance was also compared with that achieved by precious electrodes.**

According to the review article (ref. 37), Co₃O₄-based anode for OER and metal Ni-based cathode for HER have been widely applied for alkaline water electrolysis. Therefore, we employed Co₃O₄ and Ni-foam as OER and HER electrodes, respectively, to further investigate the separate steps. [In this experiment, commercialized Co₃O₄ powder was treated by ball milling for 4 hours, and then was used to fabricate OER electrode. The Co₃O₄-based OER electrode was obtained by mixing 80 wt % Co₃O₄ powder, 10 wt % Ketjen Black (KB) as conductive agent, and 10 wt % polytetrafluoroethylene (PTFE) as binder. For a typical preparation, Co₃O₄, KB, and PTFE were dissolved in isopropanol to form a slurry with the weight ratio mentioned above, and then the slurry was rolled into a film. Finally, the film was pressed on stainless steel mesh to form OER electrode (2.5×4 cm²). Commercialized Ni-foam was directly used as the HER electrode (2.5×4 cm²). An alkaline water electrolytic cell was constructed with a Co₃O₄-electrode for OER, a commercial Ni-foam electrode for HER and a Ni(OH)₂ electrode (See Movie S3 and S4 in revised manuscript).

Water electrolysis of the cell was investigated by chronopotentiometry measurements with an applied current of 200 mA. Chronopotentiometry curve (cell voltage vs. time) of the electrolytic cell is shown in **Figure answer 1a**. The chronopotentiometry curve of the cell using precious electrodes tested at the same condition is also shown in **Figure answer 1a** for comparison. In addition, OER potential on Co₃O₄ electrode and HER potential on Ni-foam electrode were investigated, in comparison with that on precious electrodes (see **Figure answer 1b**). It can be observed from **Figure answer 1a** that when using non-precious electrodes, the electrolysis process still includes two separate steps (Step 1 and 2) with different cell voltages. However, the voltages of the cell using non-precious electrodes (0.526V on step 1 and 1.611 V on step 2) are higher than that of the cell using precious electrodes (0.432V on step 1 and 1.553 V on step 2), which is owing to the lower catalytic ability of non-precious electrodes. As shown in **Figure answer 1b**, the OER potential on Co₃O₄ electrode (0.788 V vs. Hg/HgO) is higher than that on commercialized RuO₂/IrO₂ coated Ti-mesh (0.691 V vs. Hg/HgO). The HER potential on Ni-foam electrode (-1.177 V vs. Hg/HgO) is lower than that on commercialized Pt coated Ti-mesh (-1.124V vs. Hg/HgO). The cycle of

step 1 (H_2 production) and step 2 (O_2 production) in the cell using non-precious electrodes was also investigated with an applied current of 200 mA (**Figure answer 1c**). The achieved cycle performance is similar to that achieved by the cell using precious electrodes. In addition, the video evidence was also given in the revised manuscript (Movie S3, S4) to confirm the separate H_2/O_2 generation in the cell using non-precious electrodes.

Figure answer 1 | Electrochemical profile of separate H_2 and O_2 production in alkaline electrolytic cell using non-precious electrodes (Co_3O_4 for OER and Ni-foam for HER). (a) Cell voltages on step 1 and step 2 of the cells using non-precious electrodes and precious electrodes. (b) OER potentials on Co_3O_4 -based electrode and RuO_2/IrO_2 -based electrode and HER potentials on Ni-foam electrode and Pt-based electrode. (c) Chronopotentiometry curve of H_2/O_2 generation cycle with an applied current 200 mA of the cell using non-precious electrodes.

According to your kind suggestion, **Figure answer 1** and corresponding discussion have been given in the revised manuscript [please see revised manuscript (Page 10): sentences highlighted by yellow background; Please see **Figure S13** and **Movie S3-S4** for detail]

Point-3: Efficiency has been given in the revised manuscript.

According to Cronin *et al.*'s previous report (ref. 39), the efficiency of two-step system can be calculated by comparing its total driven voltage (step 1 + step 2) to the driven voltage of corresponding one-step system. In the revised manuscript, chronopotentiometry measurement with an applied current of 200 mA was employed to investigate the one-step electrolysis process that is based on a commercialized RuO_2/IrO_2 coated Ti-mesh anode ($2.5 \times 4 \text{ cm}^2$) for OER and a commercialized Pt coated Ti-mesh cathode ($2.5 \times 4 \text{ cm}^2$) for HER in an alkaline

medium. The achieved chronopotentiometry curve of one-step electrolysis is shown in **Figure answer 2a**, where it can be detected that the cell exhibits a voltage of **1.829 V** with the applied current of 200 mA. The chronopotentiometry curve tested at 200 mA of two-step system using precious electrodes [RuO₂/IrO₂ coated Ti-mesh electrode (2.5×4 cm²), Pt coated Ti-mesh electrode (2.5×4 cm²) and Ni(OH)₂ electrode (2.5×4 cm²)] is shown in **Figure answer 2b**, where it can be observed that the two steps display a total cell voltage of **1.985 V** (1.553 + 0.432 V). Therefore, the efficiency of the two-step cell using precious electrodes should be **92%** (=1.829/1.985) compared to corresponding one-step system. In addition, the one-step system that is based on non-precious electrodes [Co₃O₄-based anode (2.5×4 cm²) + Ni-foam cathode (2.5×4 cm²)] was investigated by chronopotentiometry measurement with an applied current of 200 mA (**Figure answer 2c**). It can be observed from **Figure answer 2c** that the one-step system using non-precious electrodes exhibits a cell voltage of **1.973 V**. At the same test condition, the two-step system using non-precious electrodes displays a total cell voltage of **2.137 V** (1.611 + 0.526 V) (**Figure answer 2d**). The efficiency also is about **92%** (1.973/2.137) compared to corresponding one-step system.

Figure answer 2 | Driven voltage comparison at 200 mA between one-step system and two-step system. (a) one-step electrolysis using precious electrodes. (b) two-step electrolysis using precious electrodes. (c) one-step electrolysis using non-precious electrodes. (d) two-step electrolysis using non-precious electrodes.

In our manuscript (see **Figure 3C**), the typical drainage method has been employed to measure the H₂ production rate (mL at 1atm) of the cell using precious electrodes with an applied current of 1000 mA for 100 s. It can be observed in **Figure 3C** that about **12 mL** H₂ was generated over the 100s electrolysis. According to the equation of ($I \times t = nZF$), the theoretical value should be **12.67 mL** (at 1atm) with the applied current of 1000 mA for 100 s.

Therefore, the faradaic efficiency is **94.7%** (12/12.67). In theory, the faradaic efficiency should be 100%. However, the impedance and dissolution of H₂ in aqueous solution might slightly reduce the efficiency.

According to your kind suggestion, all the data about efficiency have been shown in the main-text of revised manuscript (see revised manuscript (page 10): sentences highlighted by yellow background). In addition, **Figure answer 2** and corresponding discussion have been given in the manuscript as detailed supporting information (see **Figure S14**). Please also see the discussion about Figure 3C and 3D in revised manuscript.

Point-4: Energy density of the battery is calculated.

Figure answer 3 | Charge curve of the H₂-production process (Step 1, black line) and the discharge curve of NiOOH-Zinc battery (Step 2', red line). Inset: corresponding electrode reactions (charge current: 200 mA; discharge current: 200 mA).

As shown in **Figure answer 3**, the active materials for the charge/discharge cycle are H₂O, Ni(OH)₂ and Zn. According to these electrode reaction equations, it needs 1 mol H₂O (18g), 1 mol Ni(OH)₂ (92.7g) and 0.5 mol Zn (0.5× 65.38g) to store/deliver 1 mol electron (= 96500 C). Therefore, the energy density of the rechargeable cycle can be calculated by following equation

$$E = \frac{Q \times V}{M_{Ni(OH)_2} + M_{H_2O} + 0.5M_{Zn}} \times \frac{1000}{3600}$$

Herein, E is the energy density (Wh kg⁻¹), Q is the quantity of electricity (96500C), V is the average discharge voltage of the cell (1.5V), $M_{Ni(OH)_2}$ is the molecular mass of Ni(OH)₂ (92.7g), M_{H_2O} is the molecular mass of H₂O (18g) and M_{Zn} is the molecular mass of Zinc (65.38g). Thus, the calculated energy density can reach 280 Wh kg⁻¹.

This theoretical energy density is close to the theoretical energy density of conventional Ni-MH batteries, Ni-Cd batteries or Ni-Zn batteries, and is higher than the theoretical energy density of lead-acid batteries and aqueous Li-ion batteries (ref. 52).

Figure answer 3 and corresponding discussion have been given in the revised manuscript. Please see revised manuscript (page: 14): these sentences highlighted by yellow background and **Figure S19** with corresponding

discussion.

Response to Reviewer #2

Comment: Chen *et al.* discuss an intriguing concept of combining an electrolyzer electrode with a Ni(OH)₂ battery electrode to generate hydrogen gas without oxygen evolution. Instead, the battery material is charged. Using DEMS, it is demonstrated that hydrogen gas is indeed formed and that the charged battery material can either be used to generate oxygen (using a suitable electrolyzer anode) or electricity (using a Zn cathode). The idea of separating hydrogen and oxygen evolution was pioneered by Cronin, Symes and coworkers (as also cited in the manuscript). Chen *et al.* applied it successfully to more conventional electrochemical system where commercial electrodes were employed and a simple alkaline electrolyte. The combination of an electrolyzer anode with a Zn cathode is a welcome further (and original) application of the concept. As such, the manuscript should attract considerable attention in energy research. Therefore, the manuscript is suitable for Nature Communications. However, several points need to be addressed before publication can be recommended:

Response: Thank you very much for kindly reviewing our manuscript and giving a positive comment. For each question pointed out by you, we would like to answer separately and revise our manuscript according to your suggestions [In addition, according to reviewer #1,3's suggestions, some new data are given in the revised manuscript. Please also see our response to reviewer #1 (Figures answer 1-3) and to reviewer # 3 (Figures answer 11, 12)]

Question 1: The charge/discharge times of the battery electrode are 10 min.

- a) This seems too short to buffer fluctuations from renewable sources, particular the day/night cycle mentioned several times. Please comment on this for the reader.
- b) Does the device still work efficiently if the battery material is only charged partially (e.g. 1/4, 1/2, 3/4) and discharged partially (e.g. same fractions)? This mode of operation seems highly relevant to the proposed coupling to renewables with arbitrary fluctuations.

Response: Thank you very much for your very good question. The purpose for using such short time is to emphasize that we can flexibly change the operation of our system even within very short time, which is quite difficult for conventional alkaline water electrolytic cells. In fact, the operating time on each step can be adjusted easily. To clarify this point, charge/discharge time of 12 hours with an applied current of 20 mA was employed to investigate the separate H₂ production step and O₂ production step (see Figure answer 4).

Figure answer 4 | Electrochemical profile of the new type alkaline water electrolytic cell with a step-time of 12 hours and an applied current of 20 mA.[Cell structure: Pt coated Ti-mesh electrode ($2.5 \times 4 \text{ cm}^2$) for HER / $\text{Ni}(\text{OH})_2$ electrode ($2.5 \times 4 \text{ cm}^2$) / $\text{RuO}_2/\text{IrO}_2$ coated Ti-mesh electrode ($2.5 \times 4 \text{ cm}^2$) for OER]

As shown in **Figure answer 4**, the electrolytic cell can be cycled with a step-time of 12 hours, which is corresponding to the daytime and nighttime. Higher operating current with such step-time (12 hours) can also be easily achieved by increasing the amount of $\text{Ni}(\text{OH})_2$ electrode.

On the other hand, it is also well known that commercialized batteries can be cycled with different charge depths. **As a typical battery electrode, it is undoubted that $\text{Ni}(\text{OH})_2$ electrode can also be cycled with different charge depths.** To clarify this point, electrochemical profile of a commercialized $\text{Ni}(\text{OH})_2$ electrode ($2.5 \times 4 \text{ cm}^2$) was investigated with different charge depths through a typical three-electrode system [Work electrode: $\text{Ni}(\text{OH})_2$ electrode; Counter electrode: Pt coated Ti-mesh ($2.5 \times 4 \text{ cm}^2$); Reference electrode: Hg/HgO; Electrolyte: 1 M KOH]. As shown in **Figure answer 5a**, the $\text{Ni}(\text{OH})_2$ electrode was charged with an applied current of 100 mA for 3 hours to reach the full charge depth, and then the electrode was discharged to 0 V (vs. Hg/HgO) with a discharge current of 100 mA. It can be detected that the corresponding discharge time (**Figure answer 5a**) is 3 hours (= charge time), indicating a reversible cycle. When the charge depths are controlled at 75% (2.25 hours; **Figure answer 5b**), 50% (1.5 hours; **Figure answer 5c**) and 25% (0.75 hour; **Figure answer 5d**), highly reversible charge/discharge profiles still can be observed, clearly. **The results shown in Figure answer 5 demonstrate that $\text{Ni}(\text{OH})_2$ electrode can work well at various charge depths.**

Figure answer 5 | Electrochemical profile of a Ni(OH)₂ electrode (2.5×4 cm²) at different charge depths tested with three-electrode method at a charge/discharge current of 100 mA. [Work electrode: Ni(OH)₂ electrode; Counter electrode: Pt coated Ti-mesh (2.5×4 cm²); Reference electrode: Hg/HgO; Electrolyte: 1 M KOH].

Obviously, we can easily control the step-time for H₂ production and O₂ production in the new type alkaline electrolytic cell by adjusting the charge depths of Ni(OH)₂ electrode. To clarify this point, charge-time of 3 hours, 2.25 hours, 1.5 hours and 0.75 hours of Ni(OH)₂ electrode with an applied current of 100 mA were employed to control the step-time for H₂ production and O₂ production in alkaline electrolysis electrolytic cell (**Figure answer 6**). It can be observed from **Figure answer 6** that the new type alkaline electrolytic cell still can work efficiently.

According to your kind suggestion, **Figure answer 4-6** and corresponding discussion have been added in the revised manuscript. Please see revised manuscript (Page, 9-10): sentences highlighted by green background. Please also see supporting information in revised manuscript (**Figure S10-S12**)

Figure answer 6 | Electrochemical profile of the new type alkaline water electrolytic cell with different step-time and an applied current of 100 mA. (a) 3 hours, (b) 2.25 hours, (c) 1.5 hours and (d) 0.75 hour. [Cell structure: Pt coated Ti-mesh electrode ($2.5 \times 4 \text{ cm}^2$) for HER/ $\text{Ni}(\text{OH})_2$ electrode ($2.5 \times 4 \text{ cm}^2$) / $\text{RuO}_2/\text{IrO}_2$ coated Ti-mesh electrode ($2.5 \times 4 \text{ cm}^2$) for OER].

Question 2: The authors mention self-discharge of NiOOH as the reason for the current difference between hydrogen and oxygen evolution during DEMS.

a) This would be serious problem for the envisioned application. Please comment on this, preferably also in the discussion section of the manuscript.

b) Self discharge also means that the device could be operated during the day and self discharge overnight? How long would it take to self-discharge?

Response: Thank you very much for your very good question. Self-discharge is a very common phenomenon for batteries. A very limited self-discharge is a quality index of any successfully commercialized battery. Nickel hydroxide-based rechargeable batteries (such as nickel metal hydride (Ni-MH) batteries and nickel-cadmium (Ni-Cd) batteries) have been commercialized for a very long time, and still play an important role on current battery market. **Therefore, it is undoubted that the self-discharge of nickel hydroxide electrode is very limited.** We believe that a lot of readers have the experience that their full charged Ni-MH or Ni-Cd batteries can still work even after

several weeks (or months) rest. As correctly pointed out by you, we ever mentioned that there is slight self-discharge of charged Ni(OH)₂ electrode (= NiOOH electrode) in the DEMS investigation, because of the rest step of 130 minutes between H₂ production step (step 1) and O₂ production step (step 2). As shown in Figure 3C, the step-time of O₂ production is about 1700s that is close to the H₂ production (1800s). This description may lead to a misunderstanding that “the self-discharge of NiOOH electrode will be greatly aggravated with longer rest time”. To clarify this point, self-discharge performance of a NiOOH electrode (2.5×4 cm²) was investigated with three-electrode method. Firstly, the Ni(OH)₂ electrode was cycled with an applied current of 100 mA without any rest time between charge and discharge (Figure answer 7a). In the experiment, the Ni(OH)₂ was charged for 2.25 hours, and then was directly discharged to 0 V (vs. Hg/HgO). As shown in Figure answer 7a, the discharge time is 2.25 hours, indicating totally reversibility. Next, the same Ni(OH)₂ electrode was cycled at the current of 100 mA with a consecutive three-step, including a charge step of 2.25 hours, a rest step of 24 hours and a discharge step. Figure answer 7b shows the charge curve of the Ni(OH)₂ electrode over 2.25 hours. Figure answer 7c shows the potential (vs. Hg/HgO) change of the charged Ni(OH)₂ electrode over rest of 24 hours, where it can be detected that the potential reduces slightly. The discharge curve after 24 hours rest of the electrode is given in Figure answer 7d. As shown in the Figure answer 7d, the electrode after 24 hours rest still can be discharged for 2.194 hours, which is close to the discharge time of the electrode without any rest (2.25 hours; see Figure answer 7a), indicating a very limited self-discharge.

Figure answer 7 | Self-discharge profile of charged Ni(OH)₂ electrode tested with three-electrode method [Work electrode: Ni(OH)₂ electrode; Counter electrode: Pt coated Ti-mesh (2.5×4 cm²); Reference electrode: Hg/HgO; Electrolyte: 1 M KOH]. (a) Charge/discharge curve at a current of 100 mA of the electrode without rest [In this experiment, the Ni(OH)₂ was charged for 2.25 hours, and then was directly discharged to 0 V (vs. Hg/HgO)]. (b, c, d) the same Ni(OH)₂ electrode was cycled at the current of 100 mA with a consecutive three-step, including a charge step of 2.25 hours (b), a rest step of 24 hours (c) and a discharge step(d).

According to your kind suggestion, Figure answer 7 and corresponding discussion have been added in the revised

manuscript as an extend discussion for popular readers. [Please see revised manuscript (page 12): sentences highlighted by green background and Figure S16].

Question 3: P10. "The data of DEMS (i.e. ion current vs. time) can not be directly employed to evaluate the amount of H₂ and O₂."

a) Why did your engineer not recommend to perform a calibration for these two gases? See, e.g. <http://mediatum.ub.tum.de/doc/1007122/1007122.pdf>. It would strengthen the manuscript if the authors can show that the ratio between H₂ and O₂ at the end of the experiments is 2:1 as expected and observed in traditional electrolysis.

b) The following sentence states that 12 mL H₂ was produced. How was this obtained if not from DEMS? Please include the calculation in the Supplemental Document. What is the expected theoretical value for this calculation?

Response:

To a): Thank you very much for your good question. In our experiment, the coupled TA-MS system (NETZSCH, TG-209F1/QMS403C) was employed to in-situ analyze the purity of H₂ production and O₂ production over the electrolysis process. This instrument can be used to directly analyze purity of gases, especially for these gases with low molecular mass. However, the instrument cannot be directly employed to make quantification of gases. Only after upgraded with an additional device [NETZSCH, PTA 300 (Pulse-Thermo-Analysis Device)], this instrument can be used to quantify the gases. Standard gas with a specific concentration must be quantified as a reference. Then, quantification of gas can be evaluated through the TA-MS system connected with PTA 300 [said by Customer Service engineer: Chenggen Ye, E-mail: chenggen.ye@netzsch.com, Phone: +86 21 51089255-685]. However, it may take a very long time to upgrade our devices. **In addition, a typical drainage method was employed to measure the volume of generated H₂ and O₂, respectively. The achieved result indicates that the H₂/O₂ ratio is 2:1 in the consecutive cycle of step 1 and step 2 (this point will be further discussed later; see response to b) for detail).**

On the other hand, Coulombic efficiency (Q_1/Q_2) can be directly used to evaluate the ratio between H₂ and O₂ at the end of the experiments, where Q_1 is the quantity of electricity consumed on step 1 and Q_2 is the quantity of electricity consumed on step 2. The amount of generated H₂ and O₂ can be calculated by the following equations.

$$Q_1 = I_1 \times t_1 = n_{H_2} Z_1 F \quad \text{and} \quad Q_2 = I_2 \times t_2 = n_{O_2} Z_2 F$$

Herein, I_1 or I_2 is the applied current for H₂ generation or O₂ generation, n_{H_2} or n_{O_2} is the mol-value of generated H₂ or O₂ gas, t_1 or t_2 is the electrolysis time for H₂ generation or O₂ generation, Z_1 is 2 for H₂ production and Z_2 is 4 for O₂ production. For our case, a same current was applied for both steps ($I_1 = I_2$). Accordingly, the ratio between generated H₂ and O₂ can be summarized as:

$$\frac{n_{H_2}}{n_{O_2}} = \frac{t_1 \times Z_2}{t_2 \times Z_1} = \frac{t_1 \times 2}{t_2}$$

For conventional one-step water electrolysis, the ratio of generated H₂ and O₂ should be 2: 1 because of the same current ($I_1 = I_2$) and the same time ($t_1 = t_2$). In our experiment of consecutive cycle of step 1 and step 2 (without interval between step 1 and step 2), the t_1 is equal to t_2 , indicating a ratio of 2:1.

However, in the DEMS investigation, the t_2 (1700 s) is slightly smaller than t_1 (1800s), which is owing to the slight self-discharge of charged Ni(OH)₂ electrode over the interval time. Therefore, in the DEMS investigation, the ratio of generated H₂ and O₂ is 2: 0.9444. However, as shown in the response to your question 2, the self-discharge is very limited even with long interval time between step 1 and step 2.

By the way, thank you very much for giving us a very important reference for DEMS investigation. We are considering upgrade our instrument in our near future investigation.

To b): Sorry, our simple description confuses you totally. The 12 mL H₂ was measured by typical drainage method. In fact, the apparatus configuration (photo) for determining volumes of gas evolution has been shown in **Figure 3c** in main-text. However, our description is too simple to understand. In order to clarify this point, a schematic illustration was given as **Figure answer 8**.

Figure answer 8 | Schematic illustration of a typical drainage device for gas volume measurement. [Photo of the device is given in Figure 3c]

As shown in **Figure answer 8**, the generated gas can be collected in tube-A. We should shift the tube-B during the measurement to keep the liquid level in tube-B be as same as that in tube-A, then we can read the volume of generated gas. In our manuscript, H₂ production volume of the cell was tested with an applied current of 1000 mA for 100 s by using above device. It can be observed in **Figure 3C** that about **12 mL** H₂ was generated over the 100s electrolysis. According to the equation of ($I \times t = nZF$), the theoretical value should be **12.67 mL** (at 1atm) with the applied current of 1000 mA for 100 s. Therefore, the faradaic efficiency is **94.7% (12/12.67)**. In theory, the faradaic efficiency should be 100%. However, the impedance and dissolution of H₂ in aqueous solution might slightly reduce the efficiency. In the revised manuscript, the same method was employed to measure the volume of O₂ generated over step 2 at the same current of 1000 mA. The achieved result indicates that about **6 mL** O₂ was generated over step 2. Therefore, the ratio of H₂ and O₂ is 2:1 in the consecutive cycle of step 1 and step 2.

According to your kind question, the gas volume measurement was further described in revised manuscript. **Figure answer 8** is also given in supporting information as **Figure S17** to clarify the gas volume measurement. Furthermore, the faradaic efficiency (94.7 %) and the ratio of H₂ and O₂ (2:1) are also mentioned in the revised

manuscript [please see revised manuscript (page 12): sentences highlighted by green background.]

Question 4: For H_2/O_2 operation, 20 cycles are shown in in Fig. 2 and for $H_2/Zn(OH)_2$ operation, 5 cycles. Were the operating modes tested by cycling further? These cycle numbers are too low to estimate usefulness and claim the battery is "rechargeable".

Response: 100 cycles of H_2/O_2 operation are given as **Figure answer 9**, which has been added in our revised manuscript. [See: revised manuscript (page 9): sentences highlighted by green background and **Fig. S9** in supporting information]

Figure answer 9 | Electrochemical profile of 100 cycles of H_2/O_2 operation [Test current: 200 mA; step-time: 600 s; cell structure: Pt coated Ti-mesh electrode ($2.5 \times 4 \text{ cm}^2$) for HER / $Ni(OH)_2$ electrode ($2.5 \times 4 \text{ cm}^2$) / RuO_2/IrO_2 coated Ti-mesh electrode ($2.5 \times 4 \text{ cm}^2$) for OER].]

In addition, 50 cycles of $H_2/Zn(OH)_2$ operation are shown in **Figure answer 10**, which has been added in the revised manuscript.[See: revised manuscript (page 13): sentence highlighted by green background and **Fig. S18** in supporting information]

Figure answer 10 | Electrochemical profile of 50 cycles of H_2 production (step 1) and discharge of Zn-NiOOH battery (step 2') [Test current: 200 mA; charge-time: 600 s; Cell structure: Pt coated Ti-mesh electrode (2.5×4

cm²) for HER / Ni(OH)₂ electrode (2.5 × 4 cm²) / Zn-anode (2.5 × 4 cm²)

Question 5: How does the authors work compare to previous work of the Cronin group? Is it possible to compare efficiencies?

Response: Thank you very much for your good question. The answer is summarized as:

- (1) It should be the first time that conventional alkaline water electrolysis was split into separate H₂ production and O₂ production steps by using solid-state Ni(OH)₂/NiOOH electrode as a recyclable redox mediator. Cronin *et al.*'s redox mediator (H₃PMo₁₂O₄₀) can not be used in alkaline system.
- (2) In present work, the alkaline water electrolytic cell can separately produce H₂ and O₂ without any membrane. However, membrane is still necessary in Cronin *et al.*'s investigation to separate the dissolved H₃PMo₁₂O₄₀ and HER electrode/OER electrode.
- (3) According to Cronin *et al.*'s previous report (ref. 39), the efficiency of two-step system can be evaluated by comparing its total driven voltage (step 1 + step 2) to the driven voltage of corresponding one-step system. The achieved efficiency in Cronin *et al.*'s investigation is 79%. For our case, the efficiency of the two-step cell using precious electrodes [Pt coated Ti-mesh for HER and RuO₂/IrO₂ coated Ti-mesh for OER] is 92% (=1.829/1.985) compared to corresponding one-step system. The efficiency of the two-step system using non-precious electrodes [Co₃O₄-based electrode for OER and Ni-foam electrode for HER] also is about 92% (=1.973/2.137) compared to corresponding one-step system. [According to Reviewer#1's question, the two-step H₂/O₂ production in alkaline electrolysis using non-precious electrodes was investigated in revised manuscript, and efficiency of two-step system was also evaluated, compared to corresponding one-step system; Please see our response to reviewer#1 for detail].
- (4) The alkaline water electrolytic cell can also separately produce H₂ and O₂ with non-precious catalyst.
- (5) In present work and Cronin *et al.*'s reports, Ni(OH)₂ and H₃PMo₁₂O₄₀ were used as the "electron-coupled-proton buffer (ECPB) ref.39" in alkaline water electrolysis and acid water electrolysis, respectively. However, the molecular mass of H₃PMo₁₂O₄₀ (~1800) is much higher than that of Ni(OH)₂ (~ 92). Theoretically, when 1 mol H₂ is produced, the needed mass of H₃PMo₁₂O₄₀ is much higher than that of Ni(OH)₂.
- (6) Using Ni(OH)₂/NiOOH as the recyclable redox mediator, we can combine H₂ production and discharge of NiOOH-Zn battery to provide an interesting rechargeable cycle that can produce H₂ over charge (i.e. electrolysis in step 1) and deliver energy on discharge of NiOOH-Zn battery [See Fig. 4, Movie S5 in revised manuscript].

Question 6: The title is not very clear: a) Depending on the mode, the active material is either Ni(OH)₂ or NiOOH, while only Ni(OH)₂ is mentioned in the title. "Nickel hydroxide" might be better suited. b) H₂ production is also energy storage (in the H-H bond). c) Splitting alkaline water electrolysis is not very clear either. Separating hydrogen and oxygen evolution is a much better description.

Response: Thank you very much for your good suggestion. The title has been revised as "Separating Hydrogen and Oxygen Evolution in Alkaline Water Electrolysis Using Nickel Hydroxide".

Question 7: References

a) Why are 9 references (ref. 5-13) necessary to support the sentence "hydrogen must be produced in an efficient and sustainable manner"? Isn't this covered sufficiently by the reviews in refs. 1-8? Perhaps, the original research manuscripts (9-13) could be introduced as state-of-the-art (photo) electrocatalysts.

b) The Ni(OH)₂/NiOOH electrode is central to this manuscript, yet no reviews regarding its chemistry are cited (ref. 44 seems to discuss a device). In particular, the "Bode scheme" should be mentioned in the introduction to support the transformation from Ni(OH)₂ to NiOOH. Alternatively, XRD spectra could be provided to support the transformation.

c) There should be a suitable reference on P12 to support that proton diffusion is limiting the redox processes in Ni(OH)₂/NiOOH for non-expert readers.

Response: Thank you very much for your very kind suggestion. **Firstly**, the citation about (ref. 1-13) has been revised according to your suggestion. **Secondly**, the reversible transformation between Ni(OH)₂ and NiOOH and its application in current alkaline-electrolyte based rechargeable batteries, such as Ni-MH & Ni-Cd batteries, have been mentioned in the introduction as a "Bode scheme" (See revised manuscript (page 4): sentences highlighted by green background). **In addition**, a schematic illustration of the transformation between Ni(OH)₂ and NiOOH was also shown in the supporting information to clarify the proton de-intercalation/intercalation process over Ni(OH)₂/NiOOH conversion for layperson readers (see revised manuscript, **Figure S1**). **Thirdly**, the nickel hydroxide electrode, Ni(OH)₂/NiOOH, has been used as a positive (cathode) in commercial alkaline rechargeable batteries for more than 100 years (*J. Power Sources* 89(2000)40–45), and thus the transformation of Ni(OH)₂/NiOOH have been widely clarified (*Electrochim. Acta*.11(1966)1079; *J. Phys. Chem.* 75 (1971) 1782; *J. Power Sources* 8(1982) 229; *J. Electrochem. Soc.* 143(1996)1613). Furthermore, various technologies, such as XRD, neutron powder diffraction and spectroscopic ellipsometry, etc. have been employed to confirm the transformation of Ni(OH)₂/NiOOH (*J. Electrochem. Soc.* 136 (1989) 613; *J. Electrochem. Soc.* 45(1998)1434; *Chem. Mater.* 16(2004)3936; *Electrochim. Acta* 42(1997)1253). **Therefore**, selected references were cited to support the transformation (see revised manuscript: Ref. 43-47). **Finally**, a suitable reference has been cited to support that proton diffusion is limiting the redox process (see Ref. 53 in revised manuscript).

Question 8: Language

While the manuscript is overall understandable, the language and grammar should be improved prior to publication. Some formulations are unclear, e.g. P4: "By turning into using formed NiOOH as cathode". Is there something missing after into?

Typos (incomplete list):

Several instances (e.g. caption of Fig. 1 and the abstract): produciton -> production

P6. That using a Pt plate -> that used a Pt plate

P6. CNTs supporter -> CNT support

P9. use solar energy at noon to driven H₂-production step -> use solar energy at noon to drive H₂ production

P10. no any H₂ evolution -> no H₂ evolution

P11. Popular reader -> layperson reader

P13. Microscopes -> microscopes

P15. Solution -> Solatron

Response: Thank you very much for your kind suggestions! We have further improved the English.

Response to Reviewer #3

Comment: The submitted manuscript by Long Chen et al describes an interesting strategy of utilizing $\text{Ni}(\text{OH})_2$ as a redox mediator to decouple the HER and OER of water splitting electrolysis. This manuscript is more like an engineering article rather than a fundamental science work. Therefore, this referees has the following two primary concerns from an engineering perspective.

Response: Thank you very much for kindly reviewing our manuscript. For each question pointed out by you, we would like to answer separately and revise our manuscript according to your suggestions. [In addition, according to the suggestions of reviewers #1 and #2, some new data have been given in the revised manuscript. Please also see response to reviewer #1 (Figures answer 1-3) and to reviewer #2 (Figures answer 4-10) for detail.]

Question 1: It seems like in order to directly employ a $\text{Ni}(\text{OH})_2$ electrode as the redox mediator, all the three electrodes (HER cathode, OER anode, and $\text{Ni}(\text{OH})_2$ electrode) are immersed in the same electrolyte, therefore it is a one-compartment water electrolytic cell. Although the authors used online mass spectrometer to confirm the purity of the produced H_2 and O_2 , in practical application, it is hard to imagine purging the electrolyte constantly to remove H_2 and O_2 completely. Therefore, even if H_2 and O_2 are produced separately and consecutively, there still will be residual H_2 and/or O_2 left in the headspace of the water electrolytic cell. Hence, the potential H_2/O_2 mixing issue remains. The authors are recommended to address this point.

Response: Thank you very much for your very good question. The purpose of using one-compartment is to clarify that the gas evolution only occurs on one of electrodes (HER or OER) during Step 1 or Step 2. In fact, this problem can be addressed with “two-step alkaline water electrolysis with two separate rooms (see Figure answer 11)”, where the nickel hydroxide ($\text{Ni}(\text{OH})_2/\text{NiOOH}$) electrode is used as “a solid-state proton buffer” (please also see Fig. S1 in revised manuscript) that can be moved between room-1 for H_2 production (step 1) and room-2 for O_2 production (step 2).

Figure answer 11 | A schematic illustration of the alkaline water electrolytic with two separate rooms for H_2

production (step 1) and O₂ production (step 2), respectively. Step 1 (H₂ production) takes place in room-1, which includes an anode reaction $[\text{Ni}(\text{OH})_2 + \text{OH}^- - e^- \rightarrow \text{NiOOH} + \text{H}_2\text{O}]$ and a cathode reaction $[\text{H}_2\text{O} + e^- \rightarrow 1/2\text{H}_2 + \text{OH}^-]$. Then, the formed NiOOH electrode in room-1 is moved to room-2 for Step 2, which includes an anode reaction $[2\text{OH}^- - 2e^- \rightarrow 1/2\text{O}_2 + \text{H}_2\text{O}]$ and a cathode reaction $[\text{NiOOH} + \text{H}_2\text{O} + e^- \rightarrow \text{Ni}(\text{OH})_2 + \text{OH}^-]$.

As shown in **Figure answer 11**, the anode reaction $[\text{Ni}(\text{OH})_2 + \text{OH}^- - e^- \rightarrow \text{NiOOH} + \text{H}_2\text{O}]$ on Step 1 in room-1 releases proton (H⁺), which reacts with OH⁻ to form H₂O. At the same time, H₂O is reduced into H₂ on HER electrode (= cathode) with the reaction of $[\text{H}_2\text{O} + e^- \rightarrow 1/2\text{H}_2 + \text{OH}^-]$ in room-1. Then, the formed NiOOH electrode is moved to room-2 for O₂ production (Step 2). On Step 2, the proton is stored in NiOOH electrode through the cathode reaction of $[\text{NiOOH} + \text{H}_2\text{O} + e^- \rightarrow \text{Ni}(\text{OH})_2 + \text{OH}^-]$, and simultaneously OH⁻ is oxidized on the anode (i.e. OER electrode) through the reaction of $[2\text{OH}^- - 2e^- \rightarrow 1/2\text{O}_2 + \text{H}_2\text{O}]$. As mentioned above, the nickel hydroxide (Ni(OH)₂/NiOOH) electrode is used as “a solid-state proton buffer” that can be moved between room-1 for H₂ production (step 1) and room-2 for O₂ production (step 2).

Theoretically, the method shown in **Figure answer 11** can avoid the potential H₂/O₂ mixing issue mentioned by you. In order to further clarify this point, the H₂/O₂ production with two separate rooms was also investigated in the revised manuscript (see **Figure answer 12**). The achieved electrochemical profile is as same as that achieved by one-compartment water electrolytic cell. In addition, Movie S6 and S7 were given in revised manuscript to show the separate H₂/O₂ production with two separate rooms.

Figure answer 12 | Chronopotentiometry curve (cell voltage vs. time) of the alkaline electrolytic cell with separate H₂/O₂ production in different rooms. In this experiment, step 1 is performed in room-1 with a current of 200 mA for 600s. Then, the formed NiOOH electrode is moved to room-2 for O₂ production with the same current of 200 mA. Please also see Movie S6 and S7. [Electrodes: Pt coated Ti-mesh electrode (2.5 × 4 cm²) for HER / Ni(OH)₂ electrode (2.5 × 4 cm²) / RuO₂/IrO₂ coated Ti-mesh electrode (2.5 × 4 cm²) for OER]

The response mentioned above has been given in the discussion section of our revised manuscript (Please see revised manuscript (page 15): sentences highlighted by blue-green background). In addition, **Figure answer 11**, **Figure answer 12** and **Movies S6 and S7** have been added in the revised manuscript as supporting information (Please see **Figures S20, S21 and Movies S6 and S7**)

Question 2: the stability of the in situ generated NiOOH during HER is crucial to the success of this new design of water electrolysis. As the authors implied in the text, there is a possibility that NiOOH will self discharge. Then the questions are what the capacity of the a typical Ni(OH)₂ electrode and how long it can sustain its charge before discharge, especially in a strongly alkaline aqueous electrolyte?

Response: Thank you very much for your good question. In fact, reviewer # 2 pointed out the same question (Please see our response to question 2). For convenience, herein we repeat the response briefly.

(1) Ni(OH)₂ with a theoretical capacity of 289 mAh/g is a quite stable electrode material in alkaline electrolyte solution, and has been widely used in rechargeable batteries using alkaline electrolyte for a very long time [about 100 years; *J. Power Sources* 89(2000)40–45]. Ni(OH)₂/NiOOH transformation based rechargeable batteries, such as nickel metal hydride (Ni-MH) and nickel–cadmium (Ni-Cd) batteries, still play an important role on current battery market.

(2) As mentioned above, the capacity of Ni(OH)₂ is 289 mAh/g. For a specific electrode, the total capacity depends on the mass loading of Ni(OH)₂. For our case, the Ni(OH)₂ electrode (2.5 × 4 cm²) can be charged for 3 hours with a current of 100 mA to reach the full charge depth, indicating a capacity of 300 mAh. Especially, Ni(OH)₂ electrode can be cycled with various charge depths, and thus we can adjust the step-time for H₂ and O₂ production by controlling the charge depths of Ni(OH)₂. Please see our response to question (1) of reviewer # 2 for detailed information.

(3) Self-discharge is a very common phenomenon for batteries. A very limited self-discharge is a quality index of any successfully commercialized battery. Therefore, it is undoubted that the self-discharge of charged Ni(OH)₂ electrode (=NiOOH electrode) is very limited. We believe that a lot of readers have the experience that their full charged Ni-MH or Ni-Cd batteries still can work even after the several weeks (or months) rest.

(4) As correctly pointed out by you, we ever mentioned that there is slight self-discharge of NiOOH electrode in the DEMS investigation, because of a rest step of 130 minutes between H₂ production step (step 1) and O₂ production step (step 2). As shown in Figure 3C, the step-time of O₂ production is about 1700s that is close to the H₂ production (1800s). This description may lead to a misunderstanding that “the self-discharge of NiOOH electrode will be greatly aggravated with longer rest time”. In fact, the self-discharge of a charged Ni(OH)₂ electrode is very limited even with long time. To clarify this point, a self-discharge performance of a charged Ni(OH)₂ electrode (2.5 × 4 cm²) was investigated with a consecutive three-step, including a charge step of 2.25 hours, a rest step of 24 hours and a discharge step. The NiOOH electrode after 24 hours rest still can be discharged for 2.194 hours, which is close to the discharge time of the NiOOH electrode without any rest (2.25 hours), confirming a very limited self-discharge. (Please see Figure answer 7 in our response to question 2 of reviewer # 2 for detail).

According to your kind suggestion, Figure answer 7 and corresponding discussion have been added in the revised manuscript as an extend discussion for readers (Please see Figure S16 for detail).

Reviewers' Comments:

Reviewer #2 (Remarks to the Author)

Revision has improved the manuscript by Chen et al. Several new experiments were performed to further support the claims in the manuscript and clarify questions of the reviewers. New text passages (and supporting figures) make the author's work more easily understandable for the broad readership of Nature Communications. Discussion of novelty and potential impact of the work has also been improved after revision. My questions were answered well and all requested changes were implemented. I have no further scientific comments or suggestions and the revised manuscript could be published as is.

Reviewer #3 (Remarks to the Author)

The authors have made a thorough revision to the manuscript, which is recommended to be accepted.

Response to Reviewer # 2

Overall Comment: Revision has improved the manuscript by Chen et al. Several new experiments were performed to further support the claims in the manuscript and clarify questions of the reviewers. New text passages (and supporting figures) make the author's work more easily understandable for the broad readership of Nature Communications. Discussion of novelty and potential impact of the work has also been improved after revision. My questions were answered well and all requested changes were implemented. I have no further scientific comments or suggestions and the revised manuscript could be published as is.

Response: Thank you very much for kindly reviewing our manuscript (NCOMMS-00746A) and giving a recommendation for publication.

Response to Reviewer # 3

Overall Comment: The authors have made a thorough revision to the manuscript, which is recommended to be accepted.

Response: Thank you very much for kindly reviewing our manuscript (NCOMMS-00746A) and giving a recommendation for publication.